# Melatonin Alleviates Lipopolysaccharide-Induced Endometritis by Inhibiting the Activation of NLRP3 Inflammasome through Autophagy

**DOI:** 10.3390/ani13152449

**Published:** 2023-07-28

**Authors:** Yujin Gao, Yina Li, Jiamian Wang, Xijun Zhang, Dan Yao, Xuanpan Ding, Xingxu Zhao, Yong Zhang

**Affiliations:** 1College of Veterinary Medicine, Gansu Agricultural University, Lanzhou 730070, China; gyj1234561202@163.com; 2Gansu Key Laboratory of Animal Generational Physiology and Reproductive Regulation, Lanzhou 730070, China; lyn9097@163.com (Y.L.); 17899319313@163.com (J.W.); zhangxijun0325@163.com (X.Z.); 15776502994@163.com (D.Y.); a1910225288@163.com (X.D.)

**Keywords:** bovine endometritis, mitophagy, mtROS

## Abstract

**Simple Summary:**

Bovine endometritis is characterized by reduced milk production and high infertility rates, resulting in substantial economic losses for the dairy farming sector. Melatonin, an amine hormone produced in the mammalian pineal gland, has been widely studied for its anti-inflammatory effects. In this work, we aimed to investigate whether melatonin inhibits Lipopolysaccharide (LPS)-induced endometritis and explore its anti-inflammatory mechanism. In LPS-induced bovine endometrial epithelial cell lines (BEND cells), melatonin promotes autophagy to inhibit the NOD-like receptor family pyrin domain-containing 3 (NLRP3) inflammasome activation and thus exerts anti-inflammatory effects. In a mouse model of LPS-induced endometritis, melatonin inhibited the expression of inflammatory factors and alleviated pathological changes. These findings demonstrate that melatonin inhibition of LPS-induced inflammation in vivo and in vitro may be a novel treatment for endometritis.

**Abstract:**

Bovine endometritis is characterized by reduced milk production and high rates of infertility. Prior research has indicated that melatonin may possess anti-inflammatory and antioxidant properties that can counteract the progression of inflammatory diseases. In this research, we attempted to elucidate the protective effects of melatonin on LPS-induced endometritis. The results obtained from enzyme-linked immunosorbent assay (ELISA) and quantitative real-time PCR (qRT-PCR) revealed that melatonin effectively reduced the production and release of pro-inflammatory cytokines in an LPS-induced bovine endometrial epithelial cell line (BEND cells). Furthermore, western blotting demonstrated that melatonin treatment reduced the expression levels of the NOD-like receptor family pyrin domain-containing 3 (NLRP3) inflammasome-related proteins, including NLRP3, activated caspase-1, and cleaved IL-1β. Importantly, we further demonstrated that the anti-inflammatory effect of melatonin on BEND cells was related to autophagy by western blotting. Moreover, we used western blotting to detect autophagy-related proteins, MitoSOX to detect mitochondrial reactive oxygen species production (mtROS), and mitochondrial membrane potential (MMP) assay to detect mitochondrial membrane potential. The administration of melatonin demonstrated a significant enhancement in autophagy within BEND cells, leading to the effective elimination of impaired mitochondria. This process resulted in a reduction in the generation of reactive oxygen species within the mitochondria, restoration of mitochondrial membrane potential, and inhibition of the NLRP3 inflammasome activation. Moreover, in a mouse model of LPS-induced endometritis, melatonin treatment repressed the expression of pro-inflammatory cytokines by ELISA and qRT-PCR, alleviated pathological changes by hematoxylin–eosin staining (H&E), and inhibited myeloperoxidase (MPO) activity. In conclusion, our study showed that melatonin inhibited the activation of the NLRP3 inflammasome in BEND cells through autophagy, which may provide a novel therapeutic strategy for bovine endometritis.

## 1. Introduction

Endometritis is a prevalent reproductive disorder. In general, cows infected with endometritis are characterized by the discharge of decaying, moist, reddish-brown secretions from the uterus. When cattle are seriously infected, they can cause systemic symptoms, such as increased body temperature, increased breathing, and even toxemia [1]. Endometritis causes significant economic losses to the dairy farming industry due to the low reproductive efficiency of affected animals and, subsequently, low milk yield [2]. Clinically, although various antibacterial drugs have certain curative effects on endometritis, drug-resistant bacterial strains are increasing with the use of antibacterial drugs, and these drugs persist in milk and meat, which cause hidden dangers to human health and lead to food safety problems [3]. Therefore, understanding the pathogenesis of endometritis with the aim of identifying highly efficient, low-toxicity, residue-free drugs has become a hot spot in the field of endometritis research.

There are many factors that cause endometritis in cows, and pathogenic microorganisms are the leading cause of endometritis [4]. It is well known that *Escherichia coli* (*E. coli*) is a pathogenic bacterial species that is one of the most commonly involved in the development of bovine endometritis [5]. Lipopolysaccharide (LPS), a crucial constituent of the outer membrane in gram-negative bacteria, has been shown to induce extensive systemic inflammatory responses through circulation [6]. Toll-like receptors (TLRs) play a crucial role in innate immune responses and are expressed in both myeloid lineage cells and some non-immune cells (such as epithelial cells, endothelial cells, and fibroblasts). LPS, as a pathogen-associated molecular pattern (PAMPs), can be recognized by TLR4 to activate the downstream signaling pathways (such as NF-κB and IRF3) and ultimately promote the expression of inflammatory factors [7]. Among these inflammatory cytokines, IL-1β is critical to the host’s defensive response to infection and injury and is associated with endometritis [8,9]. Additionally, IL-1β is produced via activating the NOD-like receptor family pyrin domain-containing 3 (NLRP3) inflammasome [10]. Excessive NLRP3 inflammasome activation causes tissue damage and immune dysfunction, leading to the occurrence and development of several inflammatory diseases, such as endometritis [11,12]. Hence, inhibition of NLRP3-caspase-1-mediated IL-1β production may be a possible target for ameliorating endometritis.

Damaged mitochondria are the primary and most fundamental source of intracellular reactive oxygen species (ROS), which result in decreased mitochondrial membrane potential and, ultimately, the loss of mitochondrial function [13]. Mitochondrial ROS (mtROS) can regulate the formation of the NLRP3 inflammasome and trigger the inflammatory response by promoting NLRP3:ASC:pro-caspase-1 complex assembly [14]. Autophagy, a quality control process that clears damaged proteins and organelles, can negatively regulate NLRP3 inflammasome activation through several pathways [15]. It has been demonstrated that autophagy can remove damaged mitochondria, resulting in the decreased release of mitochondria-derived DAMPs and suppression of NLRP3 inflammasome activation [16]. Thus, the removal of damaged mitochondria by autophagy to inhibit the excessive activation of NLRP3 inflammation may be a strategy for the amelioration of endometritis.

Melatonin (N-acetyl-5-methoxytryptamine), a serotonin derivative and the main secretory product of the pineal gland, plays a pivotal role in various physiological functions, such as anti-inflammatory, antioxidative responses, immune responses, circadian rhythm, and apoptosis [17]. Accordingly, melatonin can ameliorate various diseases, including cancers, neurodegenerative diseases, and inflammation [18]. In recent years, research has confirmed that melatonin exerts anti-inflammatory effects through a variety of mechanisms. Melatonin activates IκB kinase α (IKKα) in the early stage of the inflammatory response and then binds to the NF-κB-DNA complex in the late stage to inhibit the expression of inducible nitric oxide synthase (iNOS), cyclooxygenase-2 (COX-2) and pro-inflammatory cytokines (TNF-α and IL-1β), thus inhibiting the inflammatory response [19]. Melatonin also inhibits the STAT3-induced expression of IL-6 in the colons of mice, thereby reducing the intestinal inflammatory response [20]. Since melatonin exerts an anti-inflammatory effect, we wondered if melatonin could alleviate endometritis. The objective of this study was to investigate the anti-inflammatory effects of melatonin on endometritis and elucidate its underlying molecular mechanism.

## 2. Materials and Methods

### 2.1. Cell Culture and Treatment

A bovine endometrial epithelial cell line (BEND cells) was purchased from the American Type Culture Collection (ATCC, Manassas, VA, USA). DMEM (01-172-1ACS, BI, Kibbutz Beit Haemek, Israel) supplemented with a 10% fetal bovine serum (04-001-1A, BI, Kibbutz Beit Haemek, Israel) was used to culture cells in an incubator at 37 °C with CO_2_. BEND cells were stimulated with 1 μg/mL LPS (L2637, Sigma, St. Louis, MO, USA) for 24 h. BEND cells were treated with 1 mM melatonin (M5250 Sigma, St. Louis, MO, USA) for 1 h before LPS treatment. After these treatments, the supernatant and cells were used for subsequent experiments.

### 2.2. CCK-8 Assay

BEND cells were cultured in 96-well plates, and the cells were divided into six groups according to the concentration of melatonin treated (0, 0.5, 1, 2, 4, and 8 mM). BEND cells were cultured in 96-well plates, and the cells were divided into seven groups according to the concentration of LPS treated (0, 0.5, 1, 5, 10, 50, and 100 µg/mL). After 24 h of cell culture, 10 μL of CCK-8 solution (CA1210 Solarbio, Beijing, China) was added to the cells and incubated for 3 h. The absorbance was measured at 450 nm with a microplate reader.

### 2.3. Enzyme-Linked Immunosorbent Assay (ELISA)

The cells were divided into three groups (control group, LPS group, and LPS + melatonin group) or four groups (the control group, the LPS group, the LPS + melatonin group, and the LPS + melatonin + 3-MA group) according to the indicated assays. The levels of the IL-1β, IL-6, and TNF-α proteins in cell supernatants and mouse uteri were measured using ELISA kits (Jingmei Biotechnology, Nanjing, China) in accordance with the manufacturer’s instructions.

### 2.4. Quantitative Real-Time PCR

The cells were divided into three groups (control group, LPS group, and LPS + melatonin group) or four groups (the control group, the LPS group, the LPS + melatonin group, and the LPS + melatonin + 3-MA group) according to the indicated assays. Total RNA was extracted by TRIzol reagent (G3013, Servicebio, Wuhan, China). Next, total RNA was reverse-transcribed into cDNA with an Evo M-MLV reverse transcription kit (AG11732, Accurate Biotechnology, Changsha, China). qPCR was performed using the SYBR Green Pro Taq HS Premix (AG11746, Accurate Biotechnology, Changsha, China). The mRNA levels were normalized to β-actin and calculated with the 2^−ΔΔCt^ method. The mRNA-specific primers are listed in Table 1.

### 2.5. Western Blotting

The total proteins were extracted from BEND cells with RIPA buffer (G2002, Servicebio, Wuhan, China), and the protein concentration was determined by BCA protein assay (PC0020, Solarbio, Beijing, China). A loading buffer was added to the protein samples, and the samples were denatured in a water bath at 100 °C for 10 min. Samples containing the same amount of protein (50 μg) were separated with SDS-polyacrylamide gels. Then, the separated proteins were transferred to polyvinylidene difluoride (PVDF) membranes. PVDF membranes were blocked in 5% skim milk at room temperature and shaken gently for 2 h. Then, these membranes were incubated at 4 °C overnight with primary antibodies diluted with phosphate-buffered saline-containing tween (PBST) at 1:1000. These primary antibodies included anti-NLRP3 and anti-Parkin (ab263899, ab77924, Abcam, Cambridge, MA, USA); anti-caspase-1, anti-p62, and anti-LC3 (24232, 88588, 19848, Cell Signaling Technology, Danvers, MA, USA); anti-IL-1β, anti-Beclin1, and anti-PINK1 (SRP8033, SAB1306537, P0076, Sigma, St. Louis, MO, USA); and anti-COX-IV and anti-GAPDH (bsm-60712R, bsm-33033M, Bioss, Beijing, China). These membranes were washed with PBST three times and incubated with appropriate horseradish peroxidase (HRP)-conjugated secondary antibodies at room temperature for 2 h. Protein bands were detected by a chemiluminescence detection system, and GAPDH was used as the internal standard. Raw Western blot data with molecular weight markers are presented in Appendix A.

### 2.6. Mitochondrial ROS Determination

The cells were divided into three groups (control group, LPS group, and LPS + melatonin group) or four groups (the control group, the LPS group, the LPS + melatonin group, and the LPS + melatonin + 3-MA group) according to the indicated assays. BEND cells were treated with 3-MA (5 mM) for 1 h before melatonin (1 mM, 1 h) treatment and then stimulated with LPS (1 µg/mL) for 24 h. Then, the cells were incubated with 5 μM MitoSOX (M36009, Invitrogen, Waltham, MA, USA) at 37 °C for 20 min. After washing with PBS twice, the fluorescence intensity of the cells was observed by fluorescence microscopy (APExBIO, Houston, TX, USA).

### 2.7. Mitochondrial Membrane Potential (MMP) Assay

JC-1 is a dual fluorescent dye that forms a green fluorescent monomer form at low potential and a red fluorescent aggregate form at high potential. JC-1 exists as an aggregate in the mitochondria of normal cells, showing bright red fluorescence. When the mitochondrial membrane potential is damaged in cells, JC-1 exists as a monomer, showing green fluorescence. The cells were divided into three groups (control group, LPS group, and LPS + melatonin group) or four groups (the control group, the LPS group, the LPS + melatonin group, and the LPS + melatonin + 3-MA group) according to the indicated assays. BEND cells were cultured in 6-well plates and treated with 5 mM JC-1 (C2005, Beyotime, Shanghai, China) at 37 °C for 20 min. After washing with the JC-1 dyeing buffer twice, the fluorescence intensity of the cells was observed by fluorescence microscopy (APExBIO, Houston, TX, USA). Fluorescence intensity was evaluated using Image J software 1.48v.

### 2.8. Animal Studies

Thirty BALB/c female mice (age 6–8 weeks, weight 20–25 g) were purchased from the Lanzhou University Animal Core. Before the experiment, the mice were placed in an animal house with a room temperature of 25 ± 1 °C, and the mice were allowed to eat and drink freely and to adapt to the experimental environment for one week. Thirty mice were divided into 3 groups with 10 animals in each group: the control group, LPS group, and LPS + melatonin group. In the LPS group, according to previous reports, a mouse model of LPS-induced endometritis was established [9]. Briefly, mice were anesthetized with Suimixin II injection, they were fixed in a supine position for stability, and 50 μL LPS (1 mg/mL) was injected into both sides of the uterus to induce endometritis. To ensure successful model establishment, the mice were lifted tail-side up for 30 s after the LPS injection to prevent LPS reflux. In the control group, the mice were injected with an equal volume of sterile saline. In the LPS + melatonin group, the mice were intraperitoneally injected with melatonin (20 mg/kg) and treated with 50 μL LPS (1 mg/mL) 1 h later. Twenty-four hours after LPS administration, the mice were anesthetized and euthanized, and uterine tissues were collected for follow-up experiments.

### 2.9. Histological Analysis

Bilateral uterine tissues were collected and soaked in 4% formalin for 24 h. These tissues were dehydrated, made transparent, and embedded by conventional methods to prepare paraffin sections. After hematoxylin–eosin staining (H&E, Baton Rouge, LA, USA), observe the paraffin sections under the microscope (APExBIO, Houston, TX, USA) to see whether there is bleeding in the uterus, the integrity of endometrial epithelial cells and the infiltration of inflammatory cells.

### 2.10. Myeloperoxidase (MPO) Analysis

The mouse uteri were weighed, and PBS was added according to a weight-to-volume ratio of 1:9 to make tissue homogenates with a glass homogenizer. After centrifugation, the supernatants were collected to measure MPO activity using an MPO kit (ab105136, Abcam, Cambridge, MA, USA) according to the manufacturer’s instructions.

### 2.11. Statistical Analysis

Western blotting bands analysis was performed using Image J software to obtain the gray values of bands. For all cell experiments, data are shown as the mean ± SEM (*n* = 3). All the experiments were independently repeated three times. For all animal experiments, data are shown as the mean ± SEM (*n* = 10 per group). Statistical analysis and histogram drawing were performed using GraphPad Prism 8.2. (GraphPad Software, San Diego, CA, USA). One-way analysis of variance (ANOVA) with Tukey’s post-test was used for multiple comparisons. *p* < 0.05 (*), *p* < 0.01 (**), and *p* < 0.001 (***) indicate a statistically significant difference.

## 3. Results

### 3.1. Melatonin Inhibits the Expression of Pro-Inflammatory Cytokines in LPS-Stimulated BEND Cells

To verify the role of melatonin in the survival of cells, BEND cells were treated with different concentrations of melatonin for 24 h, and then, cytotoxicity was evaluated with CCK-8 assays. The findings suggested that melatonin had no effect on cell viability at concentrations below 4 mM (*p* < 0.05) (Figure 1A). We also investigated the effect of LPS on cell viability. When the concentration of LPS was greater than 50 µg/mL, it had an impact on cell viability (*p* < 0.01) (Figure 1B). To elucidate the anti-inflammatory effect of melatonin, BEND cells were pre-treated with melatonin for 1 h and subsequently treated with LPS (1 µg/mL) for 24 h. The expression of pro-inflammatory cytokines was measured by qPCR (Figure 1C) and ELISA (Figure 1D). Our data revealed that melatonin could significantly reduce the mRNA expression and release of IL-1β, TNF-α, and IL-6 compared with the LPS group (*p* < 0.001). These results demonstrated that melatonin could inhibit LPS-induced inflammatory activation in BEND cells.

### 3.2. Melatonin Suppresses NLRP3 Inflammasome Activation in LPS-Stimulated BEND Cells

NLRP3, responsible for the maturation and secretion of IL-1β, has been proven to be related to the development of endometritis in dairy cows [9]. We showed that melatonin could inhibit IL-1β expression (Figure 1C). To further demonstrate whether melatonin inhibits the activation of the NLRP3 inflammasome, the protein expression of components of the NLRP3 inflammasome, including NLRP3, pro-IL-1β, IL-1β, ASC, caspase-1, and pro-caspase-1, was measured by western blotting. These results showed that melatonin reversed the LPS-induced increase in NLRP3 inflammatory-related protein expression (Figure 2A).

### 3.3. Melatonin Promotes Autophagy and Mitophagy and Alleviates Mitochondrial Dysfunction in LPS-Stimulated BEND Cells

Melatonin has been shown to participate in the regulation of autophagy in a variety of inflammatory diseases [21]. To investigate the potential involvement of melatonin in autophagy regulation within LPS-induced BEND cells, we assessed the expression of autophagy-related proteins. The findings indicated that LPS treatment elevated the protein levels of Beclin1 (*p* < 0.001) and LC3-II (*p* < 0.01) while reducing the protein level of p62 (*p* < 0.05) compared to the control group (Figure 3A). Interestingly, melatonin treatment resulted in increased protein levels of Beclin1 (*p* < 0.05) and LC3-II (*p* < 0.05), along with a decrease in the protein level of p62 (*p* < 0.01) when compared to the LPS treatment group. These observations demonstrate the capacity of melatonin to promote autophagy in LPS-induced BEND cells.

To better explore the effect of melatonin on mitophagy, we purified mitochondria and measured the expression of mitophagy-related proteins by western blotting. The findings of this study suggested that melatonin increased the expression levels of Beclin1 (*p* < 0.001), LC3-II (*p* < 0.001), Parkin (*p* < 0.001), and Pink 1 (*p* < 0.001) and decreased p62 (*p* < 0.05) expression in BEND cells after LPS treatment, indicating that melatonin could regulate mitophagy. However, this regulatory effect was inhibited by the autophagy inhibitor 3-MA, indicating that melatonin could regulate mitophagy through autophagy (Figure 4A).

LPS treatment increases intracellular ROS levels, and mitochondrial ROS accumulation leads to a decrease in membrane potential and eventually causes mitochondrial dysfunction [22]. To determine whether melatonin is involved in the regulation of mitochondrial ROS production, mitochondrial ROS levels were measured. The findings of this study suggested that mitochondrial ROS production was significantly increased in BEND cells after LPS treatment, and this effect was reversed by melatonin (Figure 4D). To further elucidate the effect of melatonin on mitochondrial function, JC-1 dye was used to label BEND cells, and fluorescence microscopy was used to analyze the mitochondrial membrane potential. Melatonin reversed the reduction in the JC-1 polymer: monomer ratio caused by LPS, making the cells appear red; these results indicated that melatonin could reduce the damage to mitochondrial membrane potential caused by LPS and alleviate mitochondrial dysfunction (Figure 4E).

### 3.4. Melatonin Reverses the Expression of Inflammatory Cytokines, NLRP3 Inflammasome Activation, and Mitochondrial Dysfunction through Autophagy

It has been demonstrated that autophagy inhibits the activation of the NLRP3 inflammasome through various pathways [23]. Therefore, we wondered whether melatonin inhibits NLRP3 inflammasome activation through autophagy. To explore the detailed molecular mechanism by which melatonin exerts anti-inflammatory effects through autophagy, BEND cells were treated with 3-MA (autophagy inhibitor). The inhibitory effects of melatonin on the mRNA transcription and release of IL-1β (*p* < 0.001), TNF-α (*p* < 0.001), and IL-6 (*p* < 0.001) were reversed by 3-MA (Figure 5A,B). Furthermore, melatonin reversed the LPS-induced increase in the expression of NLRP3 inflammasome-related proteins, and this effect was blocked by combination treatment with 3-MA (Figure 5C). The inhibitory effects of melatonin on LPS-induced mitochondrial ROS production (Figure 5F) and mitochondrial membrane potential damage (Figure 5G) were reversed by 3-MA.

### 3.5. Melatonin Inhibits LPS-Induced Endometritis In Vivo

To validate the in vivo anti-inflammatory effect of melatonin, an experimental mouse model of LPS-induced endometritis was established, and the mice were subjected to melatonin treatment. H&E staining showed that LPS induced significant uterine tissue damage, including hemorrhage, epithelial shedding, and inflammatory cell infiltration, which were improved by melatonin (Figure 6A). The measurement of MPO activity serves as an indicator of neutrophil infiltration. The uterine tissues in the LPS group exhibited higher MPO activity compared to the control group (*p* < 0.001) (Figure 6B). However, melatonin suppressed the increase in MOP activity caused by LPS (*p* < 0.001) (Figure 6B). In addition, we observed an increase in the mRNA expression levels of IL-1β (*p* < 0.001), TNF-α (*p* < 0.001), and IL-6 (*p* < 0.001) in the LPS group, which decreased after melatonin treatment. (Figure 6C). The expression of pro-inflammatory cytokines in uterine tissues was measured by ELISA. Following LPS treatment, the expression levels of these pro-inflammatory factors were observed to be higher compared to the control group (*p* < 0.001) (Figure 6D). However, melatonin inhibited the LPS-induced upregulation of these inflammatory factors (*p* < 0.001) (Figure 6D).

## 4. Discussion

Endometritis, a common disease in cows, is one of the main causes of infertility, and it increases the elimination rate of cows and causes substantial economic losses [3]. Antibacterial drugs have a certain therapeutic effect on endometritis, but in light of the problems associated with these antibacterial drugs, we need to find an alternative strategy for treating endometritis. The anti-inflammatory effects of melatonin, an amine hormone produced in the mammalian pineal gland, have been widely studied [24]. In this work, we aimed to investigate whether melatonin inhibits LPS-induced endometritis and explore its anti-inflammatory mechanism.

LPS treatment is commonly employed to simulate inflammation in vitro and in vivo models using cells and animals, respectively. Numerous studies have demonstrated that LPS stimulates the production of IL-6, IL-1β, and TNF-α, which are also implicated in the pathogenesis of endometritis [25,26]. It has been reported that inhibiting the expression of these inflammatory factors can alleviate endometritis. Hua ZHANG et al. demonstrated that catalpol alleviates LPS-induced endometritis by inhibiting the expression of IL-6, IL-1β, and TNF-α via the TLR4/NF-κB signaling pathway [27]. In LPS-induced endometritis in mice, alpinetin exerts its anti-inflammatory effects by inhibiting the expression of IL-6, IL-1β, and TNF-α [28]. To further demonstrate the role of melatonin in the treatment of endometritis, we established LPS-induced experimental models in BEND cells and in mice. We treated mice with endometritis by intraperitoneal injection of melatonin. The intraperitoneal injection is relatively easy to operate. At the same time, the peritoneal absorption area is large, and it is full of capillaries and lymphatics, so the intraperitoneal injection of drugs has the advantages of strong absorption capacity and fast absorption speed. Our study indicated that melatonin inhibited LPS-induced production of IL-6, IL-1β, and TNF-α both in vivo and in vitro. Furthermore, these findings align with the results reported by Hu et al., where melatonin was shown to significantly mitigate uterine injury and inhibit MPO activity [12]. However, the major limitation of the present study is that we did not consider the effects of melatonin alone on mice. Future research should be undertaken to explore the effects of melatonin alone on various functions of mice.

In recent years, the role of the NLPR3 inflammasome in endometritis has received substantial attention. In 2019, Paul et al. first reported that NLRP3 inflammasome levels were increased in dairy cows with endometritis [9]. It was also reported that the mRNA expression of NLPR3 inflammasome components (NLRP3, NEK-7, ASC, and caspase-4) was significantly elevated in endometrial epithelial cells in response to inflammatory stimuli [11]. These results are consistent with our finding that the expression of NLRP3 inflammasome-related proteins in BEND cells increased after LPS treatment. Previous studies have demonstrated that the total flavonoids from Clinopodium chinense alleviate LPS-induced endometritis by inhibiting NLRP3-mediated apoptosis [29]. MiR-223 targets to inhibit the activation of NLRP3 inflammasome in bovine endometritis, which is the mechanism of its possible treatment of endometritis [9]. Therefore, inhibition of NLRP3 inflammasome activation may be a target for the treatment of endometritis. Notably, the process of NLRP3 inflammasome activation is divided into two stages. The first stage is the production of Pro-IL-1β, pro-caspase-1, and NLRP3 by transcription and translation. In the second stage, the NLRP3 inflammasome activates and secretes active caspase-1 and IL-1β [30]. We first found that melatonin could inhibit IL-1β mRNA expression and release from LPS-treated BEND cells, which provided evidence that melatonin may be involved in the inhibition of inflammasome activation. Then, we demonstrated that melatonin inhibited the expression of NLRP3 inflammasome-associated proteins, including pro-IL-1β, pro-caspase-1, and NLRP3, and inhibited the secretion of IL-1β and caspase-1 in LPS-treated BEND cells. This suggested that melatonin inhibited both stages of NLRP3 inflammasome activation. All these findings indicated that melatonin played a protective role in endometritis by inhibiting the activation of the NLRP3 inflammasome.

Excess ROS, mainly from damaged mitochondria, are important for NLRP3 activation [31]. It has been suggested that ROS participate in the development of many inflammatory diseases [32]. Therefore, inhibiting the production of excessive ROS has been a way to inhibit inflammatory diseases. A previous study noted that miR-30a attenuates endometritis in experimental mouse models by targeting MyD88/Nox2 to reduce ROS production [33]. Liu et al. demonstrated that miR-488 attenuates the production of ROS to inhibit LPS-mediated inflammatory responses in BEND cells [34]. The accumulation of mitochondria ROS and the increase in mitochondrial membrane permeability lead to a decrease in mitochondrial membrane potential, which leads to the complete loss of mitochondrial function [35]. Damaged mitochondria further release mitochondrial ROS and mitochondrial DNA, and these signals further amplify the activation of the NLRP3 inflammasome [31,36,37]. It was reported that the treatment of monocytes and macrophages with drugs that inhibit mitochondrial complexes I and III to increase mitochondrial ROS production is the primary cause of NLRP3-dependent caspase-1 activation and increased IL-1β release [31]. Mitochondrial ROS further regulate IL-1β production and activation by activating inflammasomes through the mitogen-activated protein kinase (MAPK) and extracellular signal-regulated protein kinase 1 and 2 (ERK1/2) pathways [38]. In this study, melatonin treatment effectively decreased mitochondrial ROS generation and improved the effects of LPS treatment on mitochondrial membrane potential.

Autophagy, a housekeeping pathway and mediator of cellular homeostasis, is tightly linked to the regulation of NLRP3 inflammasome activation [15,39]. Autophagy can negatively regulate inflammasome activation and reduce the inflammatory response in various ways, such as by removing intracellular DAMPs, NLRP3 inflammasome components, and cytokines [16]. In addition, melatonin has been reported to prevent atherosclerosis progression by attenuating NLRP3 inflammasome activation by activating autophagy through the Sirt3/FOXO3a/Parkin signaling pathway [40]. Despite the therapeutic potential of melatonin, there has been no evidence linking it to endometritis via autophagy. Notably, our study showed that melatonin promoted autophagy to limit NRRP3 inflammasome activation in LPS-treated BEND cells. Interestingly, the inhibitory effect of melatonin on NRRP3 inflammasome activation was attenuated by the autophagy inhibitor 3-MA. These findings demonstrated that melatonin inhibited NRRP3 inflammasome activation through autophagy.

Mitophagy, a selective form of autophagy, clears damaged mitochondria to maintain cellular homeostasis and reduces NLRP3 activation [14,41]. Supporting this notion, a recent study showed that inhibition of autophagy with 3-MA could lead to a large accumulation of mitochondrial ROS followed by enhanced activation of the NLRP3 inflammasome, which was further reversed by ROS scavengers [39,42]. We observed that melatonin promotes mitophagy in LPS-treated BEND cells. Interestingly, in LPS-treated BEND cells, melatonin combined with the autophagy inhibitor 3-MA reduces the effects of melatonin in promoting mitochondrial autophagy, inhibiting mtROS accumulation, and restoring mitochondrial membrane potential. This evidence suggests that melatonin promotes the clearance of damaged mitochondria through mitophagy. However, whether this effect is specific to mitophagy or merely the result of enhanced autophagy remains unclear and requires further investigation. This discovery supports the conclusion that melatonin treatment stimulates autophagy, leading to enhanced removal of impaired mitochondria. Consequently, this process inhibits the activation of the NLRP3 inflammasome and exerts an anti-inflammatory effect.

## 5. Conclusions

Our study provides clear evidence that melatonin inhibits LPS-induced inflammatory damage in vitro and in vivo. Melatonin promotes autophagy to inhibit the activation of the mtROS-dependent NLRP3 inflammasome and then decreases the release of IL-1β to exert its anti-inflammatory effect. All the data suggest that melatonin could be a new and promising drug for the treatment of endometritis.

## Figures and Tables

**Figure 1 animals-13-02449-f001:**
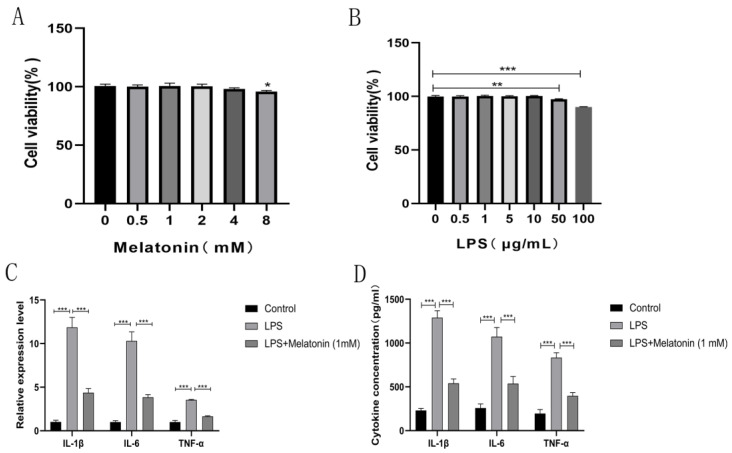
Melatonin inhibits inflammatory activation in LPS-stimulated BEND cells. (**A**) Various concentrations of melatonin (0, 0.5, 1, 2, 4, and 8 mM) were administered to BEND cells for 24 h, and cell viability was assessed using the CCK-8 assay. (**B**) Various concentrations of LPS (0, 0.5, 1, 5, 10, 50, and 100 µg/mL) were administered to BEND cells for 24 h, and cell viability was assessed using the CCK-8 assay. (**C**,**D**) BEND cells were pre-treated with melatonin (1 mM) for 1 h and subsequently co-incubated with LPS (1 µg/mL) for 24 h. The mRNA levels of IL-1β, TNF-α, and IL-6 were measured by qPCR (**C**). The secretion of IL-1β, TNF-α, and IL-6 in BEND cell supernatants was measured by ELISA (**D**). The data are shown as the means ± SEMs. All the experiments were independently repeated three times. * *p* < 0.05 melatonin (0 mM) vs. melatonin (8 mM) ** *p* < 0.01 and *** *p* < 0.001.

**Figure 2 animals-13-02449-f002:**
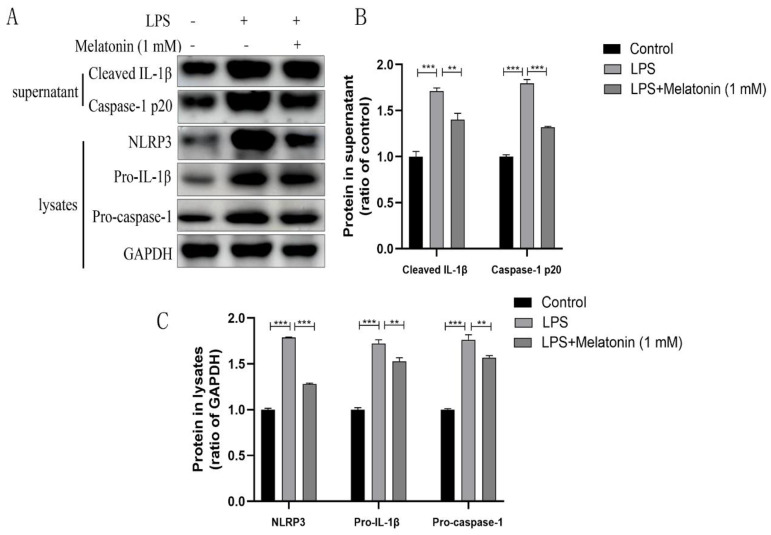
Melatonin suppresses the NLRP3 inflammasome activation in LPS-stimulated BEND cells. BEND cells were pre-treated with melatonin (1 mM) for 1 h and subsequently co-incubated with LPS (1 µg/mL) for 24 h. (**A**) The expression of cleaved caspase-1 and cleaved IL-1β in the cell supernatants was analyzed by western blotting. The expression of NLRP3, pro-caspase-1, and pro-IL-1β in cell lysates was analyzed by western blotting. (**B**,**C**) Quantification of caspase-1, cleaved IL-1β, NLRP3, pro-caspase-1, and pro-IL-1β. GAPDH was used as a control. Data were normalized to the mean value of the control group. The data are shown as the means ± SEMs. All the experiments were independently repeated three times. ** *p* < 0.01 and *** *p* < 0.001.

**Figure 3 animals-13-02449-f003:**
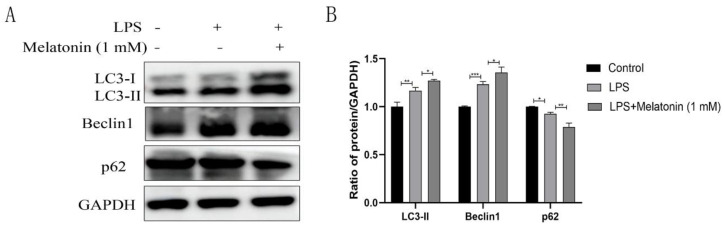
Melatonin promotes autophagy in LPS-induced BEND cells. BEND cells were pre-treated with melatonin (1 mM) for 1 h and subsequently co-incubated with LPS (1 µg/mL) for 24 h. (**A**) The expression levels of LC3, Beclin1, and p62 were measured by western blotting. (**B**) Quantification of LC3, Beclin1, and p62. GAPDH was used as a control. Data were normalized to the mean value of the control group. The data are shown as the means ± SEMs. All the experiments were independently repeated three times. * *p* < 0.05, ** *p* < 0.01, and *** *p* < 0.001.

**Figure 4 animals-13-02449-f004:**
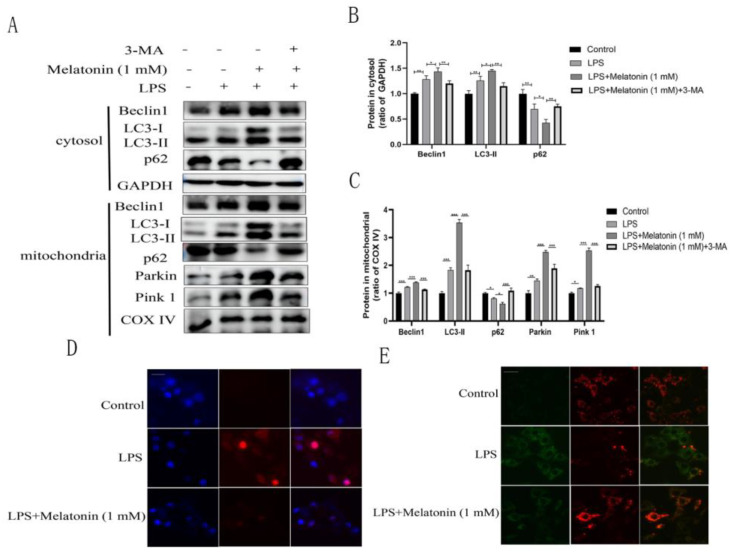
Melatonin promotes mitophagy and alleviates mitochondrial dysfunction in LPS-induced BEND cells. BEND cells were treated with or without 3-MA (5 mM) for 1 h before melatonin (1 mM, 1 h) treatment and then stimulated with LPS (1 µg/mL) for 24 h. (**A**) The expression of LC3, Beclin1, and p62 in the cell cytosol was analyzed by western blotting. The expression of LC3, Beclin1, p62, Parkin, and Pink 1 in the mitochondrial protein extracts of BEND cells was measured by western blotting. (**B**,**C**) Quantification of protein in (**A**) GAPDH was used as a control in the cell cytosol, and COX IV was used as a control in the mitochondrial. Data were normalized to the mean value of the control group. (**D**) Mitochondrial ROS levels were measured by MitoSOX. Scale bar = 20 µm. (**E**) Mitochondrial membrane potential (MMP) was analyzed by JC-1 staining. Scale bar = 20 µm. The data are shown as the means ± SEMs. All the experiments were independently repeated three times. * *p* < 0.05, ** *p* < 0.01, and *** *p* < 0.001.

**Figure 5 animals-13-02449-f005:**
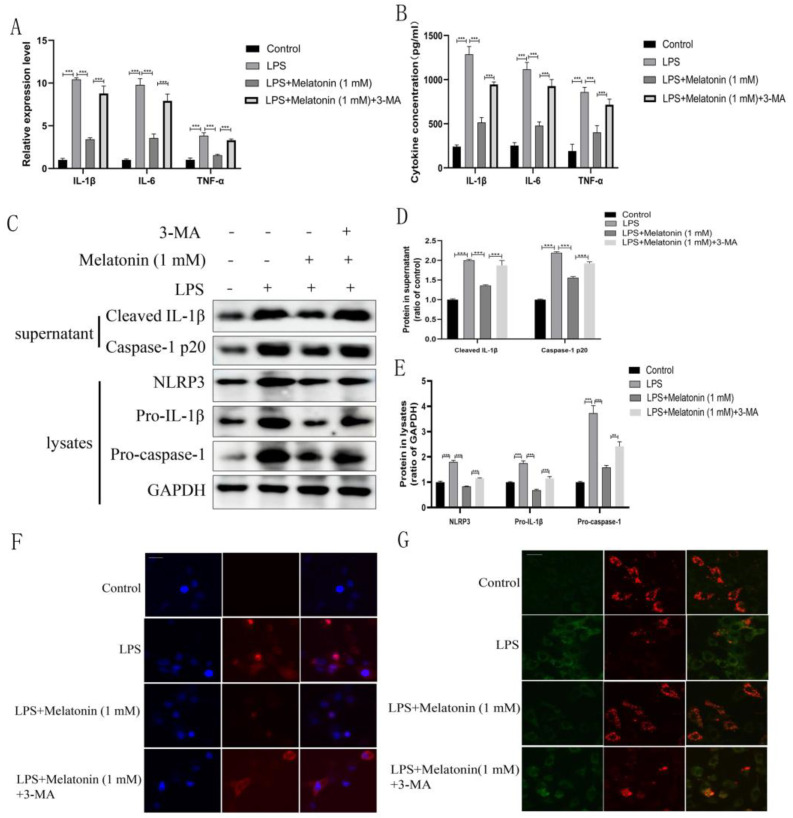
Inhibitory impact of melatonin on inflammation and mitochondrial dysfunction in LPS-induced BEND cells is counteracted by 3-methyladenine (3-MA). BEND cells were pre-treated with or without 3-MA (5 mM) for 1 h before receiving melatonin treatment (1 mM, 1 h), followed by stimulation with LPS (1 µg/mL) for 24 h. (**A**) The mRNA levels of IL-1β, TNF-α, and IL-6 were measured by qPCR. (**B**) The secretion of IL-1β, TNF-α, and IL-6 in BEND cell supernatants was measured by ELISA. (**C**) The expression of NLRP3 inflammasome-related proteins was measured by western blotting. (**D**,**E**) Quantification of caspase-1, cleaved IL-1β, NLRP3, pro-caspase-1, and pro-IL-1β. GAPDH was used as a control. Data were normalized to the mean value of the control group. (**F**) Mitochondrial ROS levels were measured by MitoSOX. Scale bar = 20 µm. (**G**) The assessment of mitochondrial membrane potential (MMP) was conducted using JC-1 staining. Scale bar = 20 µm. The data are shown as the means ± SEMs. All the experiments were independently repeated three times. ** *p* < 0.01, *** *p* < 0.001.

**Figure 6 animals-13-02449-f006:**
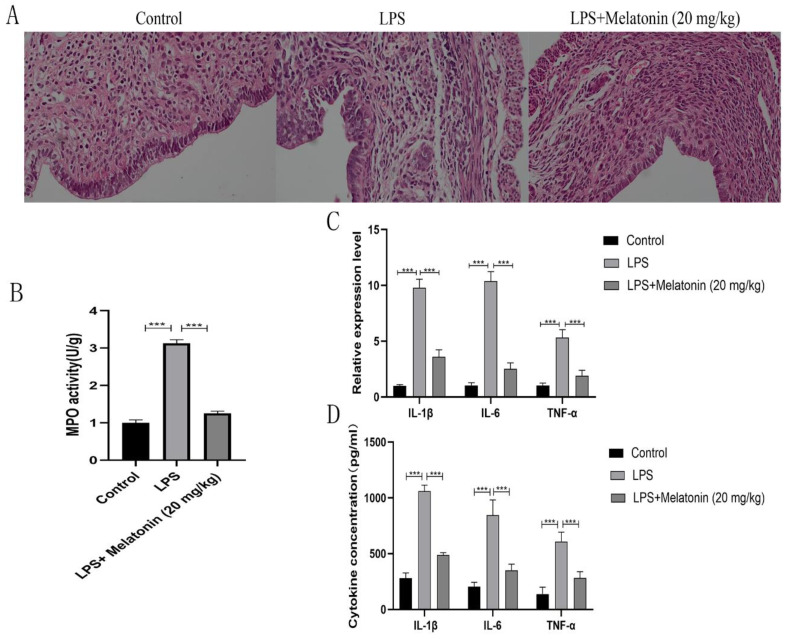
Melatonin attenuates LPS-induced endometritis in vivo. Mice were intraperitoneally injected with melatonin (20 mg/kg) and treated with 50 μL LPS (1 mg/mL) for 24 h after 1 h. (**A**) H&E staining of uterine tissues (400×). (**B**) MPO activity in uterine tissues. (**C**) The mRNA levels of IL-1β, TNF-α, and IL-6 in uterine tissues were measured by qPCR. (**D**) ELISA was used to measure the IL-1β, TNF-α, and IL-6 levels in uterine tissues. The data are shown as the means ± SEMs (*n* = 10 in each group). *** *p* < 0.001.

**Table 1 animals-13-02449-t001:** Primers design.

Species	Name	Accession Number	Primer Sequence (5′-3′)	Product Size
Bovine	IL-1β	NM_174093.1	F:AAAAATCCCTGGTGCTGGCT	195 bp
			R:GGGTGGGCGTATCACCTTTT	
	TNF-α	NM_173966.3	F:CTCCTTCCTCCTGGTTGCAG	92 bp
			R:CACCTGGGGACTGCTCTTC	
	IL-6	NM_173923.2	F:CTACCTCCAGAACGAGTATG	136 bp
			R:CAGCAGGTCAGTGTTTGTGG	
	β-actin	NM_173979.3	F:CTGTGCTGTCCCTGTATGCC	222 bp
			R:TGTCACGGACGATTTCCCGCT	
Mouse	IL-1β	NM_008361.4	F:CCTGGGCTGTCCTGATGAGAG	131 bp
			R:TCCACGGGAAAGACACAGGTA	
	TNF-α	NM_013693.3	F:CTTCTCATTCCTGCTTGTG	198 bp
			R:ACTTGGTGGTTTGCTACG	
	IL-6	NM_031168.1	F:GGCGGATCGGATGTTGTGAT	199 bp
			R:GGACCCCAGACAATCGGTTG	
	β-actin	NM_007393.5	F:AGCCATGTACGTA GCCATCC	171 bp
			R:GCTGTGGTGGTGAAGCTGTA	

## Data Availability

The data presented in this study are available upon reasonable request from the corresponding author.

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
