# Peer review of "Melatonin Alleviates Lipopolysaccharide-Induced Endometritis by Inhibiting the Activation of NLRP3 Inflammasome through Autophagy"

_animals, 2023, doi:10.3390/ani13152449_

Round 1

Reviewer 1 Report

This is a novel paper focusing on the beneficial role of melatonin in LPS-induced endometritis in vitro and in vivo study. Bovine endometritis is a serious problem for breeders and the satisfactory treatment is still missing. The paper is generally well written, concentrate on important issue. However, there are some missing information and data which at this stage evaluate on judgment of results.

Major concern:

1)      I do not see information about number of replicates for BEND cell culture and number of mice.

2)      Why only the effect of melatonin on cell viability was determined? LPS also affect the cell viability and would be beneficial to determine effect not only melatonin but also LPS and LPS and melatonin.

3)      You should also determine the effect of melatonin on all investigated parameter. It has to be shown how melatonin affect cytokine expression and concentration

4)      Please do the optical density for WB results, statistic and provide grapghs.

Minor concerns:

Add to each reagents the catalog number.

L15-16 In LPS-induced bovine (…) did you mean “In LPS treatment bovine” or “In LPS-induced endometritis”?

L44-45 I think the characteristic of endometritis should be better described.

L60-62 “LPS enters the body (…) and induce release (…) ” seems to be too much of a simplification in the publication about endometritis.

L100-106 How did you chose dose of melatonin? Based on other papers?

L174 – replace “uterine tissue” with “uteri”

L394 – replace “is a new(..) with “could”. What is your proposal to administrate melatonine to treat endometritis?

Figure 4B, 5D, and 6A – the pictures are to small

Add doses on Figures

Author Response

Dear Reviewer:

Thank you for the opportunity to revise our manuscript. We appreciate reviewers for their time and constructive suggestions. We have addressed concerns raised by reviewers point-by-point and revised our manuscript accordingly. We believe that these revisions have greatly improved our manuscript. Thank you again for your suggestions.

Reviewer 2 Report

The study describes the effect of Melatonin Alleviates Lipopolysaccharide-Induced Endometritis by Inhibiting the Activation of NLRP3 Inflammasome through Autophagy. The study was conducted correctly and concerns a fairly topical topic.

The introduction is complete and describes the different factors involved in the development of inflammation and the intervention of melatonin in mitigating the effect of ROS. Lines 92-94 should be deleted as they are superfluous to the manuscript.

M&Ms are well described and there is all the information to be able to replicate the study. Perhaps it should be better specified how many samples were treated in vitro.

The results are accompanied by various figures that can help to understand what has been achieved. However, in figures 1, 5, 6 the meaning inserted in the graphs is difficult to understand and should be improved.

The statistical analysis carried out must be better described by highlighting the variables analysed, better described.

The discussion is long and well accompanied by comparisons with other research carried out on the subject.

The result is in agreement with the aim and with the results obtained. The cited bibliography is quite extensive and perhaps some entries could be eliminated.

Ultimately the study was conducted well but some parts need to be improved before being published.

Author Response

Dear Reviewer:

Thank you very much for your kindly comments on our manuscript. There is no doubt that these comments are valuable and very helpful for revising and improving our manuscript. In what follows, we would like to answer the questions you mentioned and give detailed account of the changes made to the original manuscript.

Reviewer 3 Report

The manuscript “Melatonin Alleviates Lipopolysaccharide-Induced Endometritis by Inhibiting the Activation of NLRP3 Inflammasome 3 through Autophagy” presented exciting results and is well presented.

However, I have a few comments to be addressed before publication.

Major concern. Why did the authors decide to treat the mice with intraperitoneal melatonin and not uterine injection? 

Abstract

Please, include your methodology. It's hard to understand what was done.

L35-37 should be deleted. 

Keywords

Choose keywords that haven't been used in the title - change "Melatonin; NLRP3 inflammasome; Autophagy"

Introduction

Please review the references, some references are between extra "." (Ex. L48) others need extra space (Ex. 51).

Methodology

The tests and assays should be explained. There are no references, and replicating the study would be challenging. Please review (L100, L107, L112, L116, 143, 149 [How were the cells classified?]).

L162-167 - It needs to be clarified the experimental design. L162-3 say mice were injected with LPS, and after, say again, mice were injected with LPS after the intraperitoneal melatonin therapy.

Can the authors expand the statistical analysis? It needs to be clarified if all data was parametric or how it was tested.

Results

P values must be described for all analyses.

Portions of the results section should be moved to the discussion - Ex.: L236. Please review

Figures - Some of the graphs are too small and difficult to interpret, although nicely structured. Please, review it.

I think the discussion is nicely presented. However, it could be expanded.

Author Response

Dear Reviewer:

Thank you very much for your comments and professional advice.These opinions help to·improve academic rigor of our article. Based on your suggestion and request, we have made·corrected modifications on the revised manuscript.We hope that our work can be improved again. 

Round 2

Reviewer 1 Report

Thank you very much for improving the paper. I am satisfied almost with all corrections.

I would ask the Authors to:

- improve the sentence L63-64 – LPS is a pathogen-associated molecular pattern and induces defense-related responses. It binds to pattern recognition receptors and induces defense-related responses which are expressed on innate immune cells such as macrophages, monocytes, dendritic cells, and mast cells but also non-immune cells such as epithelial cells and fibroblasts. In this way, the innate immune response is generated. Your statement is still to general.

- include in the discussion the description of the limitation of this study. If the Authors want to prove that the treatment with melatonin would be a good treatment strategy, the treatment only with melatonin should be presented in this paper. The authors do not want to add additional experiments so in this case they should be described very carefully, that the is a lack of only melatonin treatment and this is a limitation of this study.

Author Response

(The authors gave the same response as above.)

Reviewer 3 Report

The authors have made a significant improvement. However, some points still need to be addressed.

Please, include your methodology in the abstract, as was mentioned in the previous review. There needs to be a mention even of the animal model used for your study.

Keywords - please, delete "Melatonin; NLRP3 inflammasome; Autophagy" because they are already mentioned in the Title, and replace those words.

The assays should be explained in the methodology, as mentioned previously. Replicating the study would be challenging.

Ex. "Mitochondrial Membrane Potential (MMP) Assay - how the fluorescence intensity was evaluated? What color? Etc...

Ex. Histological analysis - what pathological changes were considered?

For statistical analysis, what test was used to identify the normality of your data?

Results

P values must be described in the text.

Delete the tests used for statistical analysis from the figures' legends.

Please include "Why intraperitoneal melatonin" in the discussion and how it would be applied in clinical practice.

Author Response

Dear Reviewer:

Thank you very much for your comments and professional advice.These opinions help to·improve academic rigor of our article. Based on your suggestion and request, we have made·corrected modifications on the revised manuscript.We hope that our work can be improved again
